# The Influence of Maternal Information Sources on Infant Oral Hygiene Practices for Six-Month-Olds in South Australia: A Cross-Sectional Study

**DOI:** 10.3390/ijerph22060826

**Published:** 2025-05-23

**Authors:** Meng-Wong Taing, Wanrong Li, Loc G. Do, Diep H. Ha

**Affiliations:** 1School of Pharmacy and Pharmaceutical Sciences, The University of Queensland, Woolloongabba, QLD 4102, Australia; wanrong.li1@student.uq.edu.au; 2School of Dentistry, The University of Queensland, Herston, QLD 4006, Australia; l.do@uq.edu.au (L.G.D.); d.ha@uq.edu.au (D.H.H.)

**Keywords:** child oral health, information sources, mother–child interactions

## Abstract

This study aimed to investigate the association between the different information sources on infant oral hygiene accessed by mothers and infant oral hygiene practices in South Australia. Information on the oral hygiene practices used in 6-month-old infants—gum/tooth cleaning in the past 3 months, frequency of brushing, and the usage of fluoridated toothpaste—were acquired from the Study of Mothers’ and Infants’ Life Events Affecting Oral Health (SMILE) cohort survey. Descriptive statistics and multivariable logistic regression modelling were used to analyse the relationship between the information sources and infant oral hygiene practices. The majority of mothers (60.4%) reported not having cleaned their 6-month-old’s gums/teeth in the past 3 months. One-third of mothers with 6-month-olds did not seek information on infant oral hygiene. Mothers who sought advice from dentists were more likely to have cleaned their infant’s gums/teeth in the past 3 months, and those with infants whose teeth had erupted were also more likely to clean their infants’ teeth twice or more daily. We can conclude that mothers who sought information on infant oral hygiene from more than one source adopted generally better oral hygiene practices for their infants, with a dentist’s advice notably increasing the likelihood of mothers following the guidelines for cleaning their infants’ teeth.

## 1. Introduction

Poor oral health and hygiene—comprising dental caries, gum disease, and tooth loss—is a significant public health issue in Australia [1]. Dental caries, commonly known as tooth decay, refers to the weakening and breakdown of the tooth structure due to acid erosion of the tooth enamel, resulting in the development of cavities. Dental caries is the most prevalent form of oral disease in children, with almost half of all Australian pre-school children experiencing dental decay in their deciduous teeth [2]. The presence of dental caries in the primary dentition of young children is known as early childhood caries (ECC), which is defined as the presence of at least one decayed, missing, or filled surface on the primary set of teeth in children up to six years of age [3]. This is potentially caused by poor oral health behaviours at a young age, in particular a constant high intake of sugars, inadequate exposure to fluorides, and a lack of plaque removal through regular toothbrushing [4]. Additionally, delayed toothbrushing (beyond 2 years of age) has been identified as a risk factor for ECC, particularly in disadvantaged groups [5]. In Australia, approximately 8% of children aged 18 months old experience dental caries, and this proportion increases up to 23% by 36 months of age [6]. A higher prevalence of ECC is also noticed in Australian children who are socially or economically disadvantaged [1,5].

In 2019, the International Association of Paediatric Dentistry published an international consensus regarding the proper prevention and management of ECC, which emphasises the major role that parents and caregivers play in children’s oral health and hygiene [3]. Parents play a crucial role in shaping their children’s oral hygiene behaviour, with children being more likely to adopt regular toothbrushing routines if parents establish consistent cleaning habits [7,8]. The international consensus provided clearer guidance for the World Health Assembly, which, in 2021, approved a resolution on oral health [9]. This resolution recommended a shift in the approach to oral healthcare interventions from the traditional curative method to a preventive method that highlights the promotion of oral health within families [4,9].

Oral health knowledge is a modifiable risk factor for oral health outcomes [10], and intervening in it could foster positive dental health habits in Australian families, for both parents and their children. Within the family, emphasis has been placed on the role of the mother in relation to a child’s oral health habits and hygiene status [11]. Despite the constant shift in parental roles and responsibilities, a multivalent study by Saied-Moallemi et al. demonstrated the importance of mothers’ positive attitudes and oral health knowledge to their children’s toothbrushing behaviour and sound dentition [11]. It was also established that mothers serve as a predominant resource for their children with regard to shaping their perception and encouraging acceptance of attitudes, values, and behaviours related to oral health [11,12].

Given the limited research on the various information sources available to parents regarding children’s oral health, particularly in the context of infant oral hygiene, this study aims to determine whether the number and types of information sources consulted by South Australian mothers are associated with key oral care behaviours for their infants. These behaviours include cleaning infants’ gums or teeth and, for 6-month-olds with teeth, their frequency of teeth cleaning and use of toothpaste. By analysing these relationships, we seek to understand how diverse informational resources shape early oral hygiene practices.

## 2. Materials and Methods

### 2.1. The Study Population and Sampling Method

This report involves a data analysis of the responses obtained from the Study of Mothers’ and Infants’ Life Events Affecting Oral Health (SMILE) cohort survey dataset. SMILE was a population-based birth cohort study conducted in Adelaide, South Australia. This study implemented a multi-stage stratified sampling strategy. All newborns at the three main public hospitals in Adelaide between July 2013 and August 2014 were eligible for inclusion in the study. The data for this study were derived from the SMILE study, which consisted of multiple phases, with data being collected at various points from birth through to early childhood. This study specifically utilises the data collected when infants were six months old, focusing on the relationship between maternal information sources and infant oral hygiene practices. Unlike the broader SMILE study, which explored a wide range of health and developmental outcomes, this study narrowed its focus to oral hygiene behaviours at the six-month milestone. Recruitment was conducted by trained health professionals (dental hygienists and dental therapists) and included mothers who were both willing and able to engage in the study in both written and verbal form, ensuring an adequate level of language comprehension to complete the questionnaire. All of the information related to the methods and the data collection process is available in the report published by Do et al. [13].

### 2.2. Data Collection and Management

This study utilised data collected from questionnaires administered to the participants at the baseline and again when the infants were 6 months old. The baseline questionnaire collected data about the mother’s characteristics, which were included in the analysis to control for these confounding factors. These variables included a measure of SES, age at childbirth, indigenous status, birth country, primary language, level of education, work status, Family Tax Benefit (FTB) Part A eligibility, type of concession card, and private health insurance. The measure of SES was determined using the Index of Relative Socioeconomic Advantage and Disadvantage (IRSAD) [14] based on postal codes of residence.

A follow-up questionnaire was administered when the infants reached 6 months old and gathered information on the mothers’ management of their infants’ oral hygiene, as well as the sources of information they accessed on infant oral hygiene. The participants were asked to report the number of teeth their infants had at 6 months and to identify all relevant sources of information on infant oral hygiene. The questionnaire listed 13 options for the participants to chose from:(a)None;(b)Child health nurse;(c)Doctor/general practitioner;(d)Dietitian;(e)Dentist;(f)Mother/mother-in-law;(g)Other female relative (e.g., sister/friend with young children/playgroup mum);(h)Pamphlets from the health department/books/TV shows or videos;(i)Pharmacist;(j)Another dental professional;(k)The Internet;(l)Experience with a previous child;(m)Other.

The dependent variables and the primary focus of this study were centred on the mothers’ oral hygiene practices for their 6-month-old infants. Three variables were collected and presented as dichotomous outcomes:(a)The presence of infant gum/tooth cleaning in the last 3 months (Yes/No). Gum/tooth cleaning practices are considered to be appropriate once a baby is about 3 months of age [15]. Infant gum cleaning practices before the eruption of teeth help parents establish a cleaning routine for their infants, easing their transition to brushing once teeth appear [15].(b)The frequency of infant tooth cleaning, coded as ‘Twice or more per day’ and “Less than twice per day’. Cleaning the teeth twice daily or more is deemed to be sufficient [15,16].(c)The usage of toothpaste during infant tooth cleaning (Yes/No). The usage of fluoridated toothpaste during tooth cleaning, regardless of the amount or frequency, is considered to be inappropriate at 6 months of age [15,16].

For the purpose of this study, survey responses relating to the mother and their infant were included in the analysis; partners’ responses were not collected and therefore did not form part of the analysis. The reporting of this study conforms to the Strengthening the Reporting of Observational Studies in Epidemiology (STROBE) guidelines for cross-sectional studies by Ghaferi et al. [17].

### 2.3. The Statistical Analysis

The responses from the SMILE cohort survey dataset were descriptively analysed using IBM SPSS Statistics for Macintosh V.29.0 (IBM Corp., Armonk, NY, USA) and Microsoft Excel. The demographic characteristics of the mothers in the observed data were compared with those in the missing data, revealing no significant differences between groups. Therefore, missing data were assumed to be missing completely at random, and non-respondents were excluded from the analysis. Pearson’s χ^2^ test was used to determine significant relationships between categorical variables; the significance level was set at *p* < 0.05. When the contingency tables were larger than 2 × 2, adjusted standardised residuals was used to show cells which had larger or smaller counts than expected if two variables were considered independent.

To determine the relationship between the specific source of information accessed and whether this influenced the mothers’ oral hygiene practices for their 6-month-olds, multivariable logistic regression models were applied for each oral hygiene outcome variable while adjusting for confounding variables related to the mother using Stata software V.16.0 (StataCorp LLC, College Station, TX, USA). Bivariate logistic regression analyses were initially used to explore the unadjusted associations between each binary oral hygiene outcome variable and the independent variables. For independent variables that exhibited issues relating to separation (quasi or complete) or sparse data bias, we applied Firth’s [18] penalised maximum likelihood estimation to obtain more stable and accurate estimates [19,20]. In the multivariable adjusted logistic analyses, we applied Firth’s [18] penalisation logistic method, which improves the estimates not only for separation-causing covariates but also those for other odds ratios [20]. The multicollinearity among the independent variables was assessed. Interaction effects exploring the associations between the sources of information and the sociodemographic variables, as well as potential synergistic or contradictory combinations of effects of the information sources on the oral hygiene outcomes or demographic factors, were not possible due to the small number of cases for several information sources.

## 3. Results

### 3.1. Demographics

The characteristics of the mothers (*n* = 1479) are presented in Table 1. The majority of the mothers who responded to the survey at the baseline were aged 25–34 years old (66.0%) and were born in Australia, New Zealand or the UK (74.0%). Only a small percentage of the mothers were of Aboriginal and Torres Strait Islander origin (1.1%). Approximately one-third of the respondents had a concession card (32.7%), and half had private health insurance (50.2%), of whom 9 in 10 had dental coverage included. Slightly more than half of the mothers had completed tertiary education (51.9%), and the majority indicated that they were working full-time (38.2%) or part-time (30.1%). Regarding socioeconomic disadvantage, approximately half of the respondents lived in more disadvantaged or the most disadvantaged regions (54.1%) in South Australia.

### 3.2. The Number of Oral Hygiene Information Sources Consulted by the Mothers

The number of oral hygiene sources consulted by the mothers when their infants were 6 months of age is listed in Table 2. Of all of the respondents, 33.4% (494/1479) reported not referring to any information source on toothbrushing; 0.2% (3/1479) of the respondents did not complete the question; and 0.3% (5/1479) of the respondents indicated referring to “Other” information sources (e.g., unspecified, participated in hospital research).

The top four most reported sources of information on infant oral hygiene were (in descending order) pamphlets (359 reports); child health nurses (347 reports); other female personnel, e.g., playgroup mums (258 reports); and dentists (174 reports). From the respondents who consulted only one source (39.1%; 578/1479), most sought advice from a child health nurse (27%; 156 reports). Of the respondents who consulted two sources (16.6%; 245/1479), pamphlets (45.7%; 112 reports) and child health nurses (40.8%; 100 reports) were the most frequently accessed information sources. For the respondents consulting three sources (7.7%; 114/1479), other female personnel (66.7%; 76 reports), pamphlets (57.9%; 66 reports), and child health nurses (53.5%; 61 reports) were the preferred choices. These sources also remained the most popular when four or more sources were consulted by the respondents (3.0%; 45/1479).

### 3.3. The Association Between the Number of Oral Hygiene Information Sources Consulted by the Mothers and Infant Oral Hygiene Practices at 6 Months of Age

To understand whether the number of information sources consulted by the mothers was associated with infant oral hygiene practices, a Pearson’s χ^2^ test was performed, and the results are listed in Table 3.

The majority of the mothers (60.4%; 891/1476) reported not having cleaned their 6-month-old infant’s gums or teeth in the past 3 months. A significant association between advice-seeking behaviour and mothers having cleaned their 6-month-old’s gums/teeth in the past 3 months was observed (*p* = 0.024). Specifically, a cross-tabulation analysis identified that mothers who did not seek advice were less likely to have cleaned their 6-month-old’s gums or teeth in the past 3 months compared to mothers who had accessed one or more information sources, with the exception of mothers who had consulted four or more sources. Among the 548 mothers who reported having cleaned their infant’s gums or teeth in the past 3 months, a significant association was observed between advice-seeking behaviour and the frequency of teeth cleaning (*p* = 0.002). A trend was observed where the mothers who had accessed one or more sources were increasingly more likely to brush their infant’s teeth regularly (twice or more per day) compared to those who had not accessed any information sources. Regarding the use of toothpaste during the cleaning of the 6-month-old infant’s teeth and its association with the number of information sources accessed, no significant associations were identified. However, a trend suggested that the mothers who had accessed more sources were less likely to use fluoridated toothpaste when cleaning their infant’s teeth.

### 3.4. Influential Information Sources Enhancing Oral Hygiene at 6 Months of Age

Regarding the sources of information that increased the likelihood of a 6-month-old infant having their teeth or gums cleaned, the adjusted multivariable logistic regression model (Appendix A) showed that after accounting for maternal demographic factors, mothers who had not accessed any information sources or who relied on pamphlets/brochures were, respectively, approximately 50% and 30% less likely to have cleaned their infant’s gums or teeth in the past 3 months (OR: 0.55; 95% CI: 0.38, 0.79; *p* < 0.01 and OR: 0.71; 95% CI: 0.52, 0.96; *p* = 0.029, respectively). A similar trend was observed for mothers who sought advice from other female personnel (e.g., playgroup mothers), though these results were not statistically significant (OR: 0.73; 95% CI: 0.52, 1.02; *p* = 0.065). Conversely, mothers who sought advice from dentists were more likely to have cleaned their infant’s teeth or gums in the past 3 months (OR: 1.4; 95% CI: 0.97, 2.04), although these results were not statistically significant (*p* = 0.069). The adjusted model also revealed that certain maternal demographic factors, such as less socioeconomic disadvantage, an older age, and being born in Asia (excluding India), were significantly associated with a reduced likelihood of their 6-month-old infant having had their gums or teeth cleaned in the past 3 months (Appendix A). In contrast, the mothers whose first language was not English were significantly more likely to have cleaned their infant’s gums or teeth in the past 3 months compared to those whose first language was English (Appendix A).

In terms of the sources of information that increased the likelihood of the mothers cleaning their 6-month-old infant’s teeth twice or more per day, the adjusted multivariable logistic model (Appendix A) showed that mothers who sought advice from dentists were about three times more likely to clean their infant’s teeth twice or more per day, in accordance with the guidelines (OR: 3.2; 95% CI: 1.06, 9.4; *p* = 0.039).

Regarding the sources of information that increased the likelihood of mothers occasionally or regularly using toothpaste to brush their 6-month-old infant’s teeth, the adjusted multivariable logistic model (Appendix A) showed no significant relationship between the information sources consulted and toothpaste use. The adjusted model, however, showed that certain maternal demographic factors, such as less social disadvantage and having a university-level education or a part-time working status, were significantly associated with a reduced likelihood of occasionally or regularly using toothpaste to clean their 6-month-old infant’s teeth (Appendix A).

## 4. Discussion

This study investigated the relationship between the number and types of sources of information on infant oral hygiene consulted by South Australian mothers and its influence on oral hygiene practices in infants. A previous study by Ha and Do [21] explored the impact of early dental visits for oral health advice on reducing mothers’ unfavourable behaviours, such as putting their child to bed with a bottle. This study builds on this research by examining how the number and types of oral hygiene information sources consulted by South Australian mothers are associated with key oral care behaviours in their infants. To the authors’ knowledge, this is the first study to specifically examine this relationship using data obtained from the largest population-based oral health birth cohort study conducted in Adelaide, South Australia.

This study showed that mothers who sought advice from one or more information sources were more likely to have cleaned their 6-month-old’s gums or teeth in the past 3 months compared to those who had not accessed any information sources. However, a discrepancy was found where the mothers who accessed four or more information sources reported lower gum and tooth cleaning rates than these rates in mothers who had sought out no sources. The authors are unable to explain this anomaly, but this may in part be due to information overload, where too much advice leads to poor recall. This tendency towards information overload may be exacerbated by young dental professionals, who often feel pressured to deliver all relevant oral health education in a single appointment, fearing anything less may be seen as substandard practice [22]. This study found that one-third of the mothers with 6-month-old infants did not seek information on infant oral hygiene and were less likely to have cleaned their infants’ gums or teeth in the past 3 months. Interestingly, the mothers who referred to pamphlets or brochures were also less likely to do so. In health promotion, traditional media sources such as pamphlets and leaflets continue to be widely used to educate the public, even in today’s digital era [23,24]. The advantages of the pamphlet as an informative tool are its economic convenience and ease of circulation. However, the poor infant oral hygiene practices associated with pamphlet use by the mothers may be due to the potential ineffectiveness of print materials as tools for public health education [25]. The inappropriate levels of linguistic expression used in pamphlets may contribute to their limited effectiveness [24]. A review published in *The Health Promotion Journal of Australia* highlighted the limited effectiveness of pamphlets, where the use of pamphlets was more commonly associated with changes in knowledge and attitudes rather than health behaviour [25]. Specifically, where leaflets were used to promote oral health, a systematic review by Kay et al. found no evidence regarding an impact of oral health promotion through leaflets on oral health outcomes [26].

This study showed that mothers who sought advice from dentists were more likely to have brushed their infant’s teeth at least twice daily, in accordance with the published guidelines [15,16]. To the authors’ knowledge, this is the first study to report that seeking oral health advice from dentists, as opposed to other information sources, is associated with improved oral hygiene practices for 6-month-old infants. Studies in adult patients suggest that routine dental visits are associated with positive impacts on oral health and hygiene, with less caries experienced and better perceived oral health [27]. In addition, strong dental self-efficacy perceptions, such as a higher frequency of toothbrushing, have been found to be correlated with the provision of oral health information during dental visits [28]. The existing literature also reports an indirect effect of maternal dental attendance patterns on children’s oral hygiene behaviours [28].

There was no significant association between the information sources consulted and the use of toothpaste in 6-month-old children. Fluoridated toothpaste is not recommended for infants under 18 months [15,16] due to the risk of fluorosis, particularly between 15 and 30 months, when the primary teeth are most susceptible [29]. Interestingly, this study showed that mothers with a university-level education or working part-time were less likely to use toothpaste to clean their 6-month-old infant’s teeth. Folayan et al.’s scoping review and related studies found that higher maternal education levels are linked to better oral hygiene behaviours in children and lower risks of ECC [3,6,30]. Future efforts should prioritise educating mothers about the proper fluoride use in infants, which would particularly benefit those without a higher education and mothers in full-time work. This may promote better oral hygiene and possibly reduce the risk dental fluorosis in younger children.

Understanding the sources of information that mothers access regarding infant oral health requires consideration of the role of non-dental allied health professionals, who generally lack formal training in this area. As early oral hygiene education is not a standard component of their professional training, it is unsurprising that no significant associations were identified between their guidance and infant oral hygiene practices. In Australia, for example, pharmacists are frequently consulted on oral health issues in community pharmacy settings, and many express interest in expanding their scope of practice to support their clients better [31,32,33,34]. However, both undergraduate and postgraduate pharmacy education currently provides limited training to prepare them for this role. Recognising this gap, national collaborations such as the Australian Centre for the Integration of Oral Health have established a platform for research collaborations across the country to engage in integrated oral health initiatives. These efforts include the development of oral health training modules for non-dental health professionals, enabling an expanded scope of practice to support diverse population groups. Australian interprofessional training courses already exist for some non-dental professionals, including midwives in “Midwifery Initiated Oral Health”, which positions midwives to offer preventative oral health services involving screening, education, and referral during the prenatal period [35]. As these training programs continue to evolve and be integrated into professional education, future research should assess their effectiveness in enhancing non-dental health professionals’ capacity to influence oral health outcomes in the community.

One limitation of this study is that all of the demographic and behavioural data collected were self-reported, which may have introduced recall, social desirability, and selection biases. To mitigate memory bias, the authors designed the questionnaire items with recall periods that were reasonably short and achievable (e.g., within the last three months) to enhance the accuracy of participant recall. Another limitation of this study is the lack of assessment regarding the quality, depth, and clarity of the information resources received by the mothers, which may have led to underestimation or overestimation of the associations between behaviours and health outcomes. The statistical tests were also not powered to see differences, particularly where the sample size was small due to missing data. Future large-scale studies are needed to explore additional factors such as personal beliefs, cultural practices, and access to dental care to understand what influences infant oral hygiene outcomes better. Nonetheless, this study is the first and one of the largest international longitudinal studies on child oral health, with the sample recruitment surpassing that in other independent birth cohort studies in dental health [13]. Given that the socioeconomic characteristics of the Australian population are broadly consistent with those of the South Australians recruited in this study [36], the findings of this study may tentatively be generalised to other Australian states and territories. Nonetheless, further research is required to validate these findings. This study can build on the current findings to help reverse the trend of poor oral health in Australian children.

## 5. Conclusions

Mothers who consult multiple information sources, particularly dental professionals, are more likely to follow positive oral hygiene practices for their 6-month-old infants, such as cleaning their gums or teeth and brushing the teeth twice daily in children with primary teeth. In contrast, mothers who relied on pamphlets and brochures or who did not seek information were less likely to have cleaned their infants’ gums or teeth in the preceding three months. No relationship was found between the information sources and the use of toothpaste in infants. These findings underscore the importance of reliable, targeted information in promoting effective infant oral hygiene practices. Until non-dental health practitioners are adequately trained in oral healthcare, public health efforts should prioritise improving access to dentists. Additionally, initiatives should focus on encouraging mothers to seek oral health advice from dentists to support the optimal infant oral hygiene practices.

## Figures and Tables

**Table 1 ijerph-22-00826-t001:** Characteristics of mothers who participated in the study at 6 months.

Respondent Characteristics	Mothers (*n* = 1479)
IRSAD	
Deciles 1–2 (the most disadvantaged)	21.2% (313)
Deciles 3–5 (more disadvantaged)	32.9% (487)
Deciles 6–8 (less disadvantaged)	25.9% (383)
Deciles 9–10 (the least disadvantaged)	18.4% (272)
Not completed †	1.6% (24)
FTB Part A eligibility	
Yes	54.9% (812)
No	15.3% (227)
Unsure	28.7% (424)
Not completed †	1.1% (16)
Private insurance	
Yes	50.2% (743)
Dental coverage	
Yes	90.6% (673)
No	9.2% (68)
Not completed †	0.2% (2)
No	47.8% (706)
Not completed †	2.0% (30)
Type of concession card	
Healthcare	25.3% (375)
None	58.7% (868)
Other	7.4% (109)
Not completed †	8.6% (127)
The mother’s age at childbirth	
≤24 years	12.6% (187)
25–34 years	66.0% (975)
35+ years	20.6% (305)
Not completed †	0.8% (12)
The mother’s birth country	
Australia, New Zealand, and the UK	74.0% (1094)
Asia—other	11.0% (162)
Asia—India	7.7% (114)
Other	6.0% (89)
Not completed †	1.3% (20)
The mother’s primary language	
English	76.5% (1132)
Other	21.8% (322)
Not completed †	1.7% (25)
The mother’s indigenous status	
Yes	1.1% (17)
No	97.4% (1441)
Not completed †	1.5% (21)
Maternal education completed	
School/vocational	47.1% (696)
Some university and above	51.9% (768)
Not completed †	1.0% (15)
The mother’s work status	
Full-time	38.2% (565)
Part-time	30.1% (445)
Self-employed	3.8% (57)
Unemployed/home duties/pensioner	26.6% (393)
Not completed †	1.3% (19)

† indicates respondents who did not answer the question.

**Table 2 ijerph-22-00826-t002:** Number of reported information sources mothers consulted for oral hygiene information for infants at 6 months of age (*n* = 982).

	1 Source (*n* = 578)		2 Sources (*n* = 245)		3 Sources (*n* = 114)		4+ Sources (*n* = 45)	
Child Health Nurse	27.0%	(156)	40.8%	(100)	53.5%	(61)	66.7%	(30)
Doctor	9.2%	(53)	18.8%	(46)	28.1%	(32)	57.8%	(26)
Dietician	0.7%	(4)	0.0%	(0)	2.6%	(3)	2.2%	(1)
Dentist	12.3%	(71)	21.6%	(53)	28.9%	(33)	37.8%	(17)
Pharmacist	0.3%	(2)	0.4%	(1)	0.0%	(0)	0.0%	(0)
Other Dental Professional	0.3%	(2)	0.8%	(2)	0.0%	(0)	0.0%	(0)
Mother (-in-Law)	8.7%	(50)	24.9%	(61)	43.9%	(50)	66.7%	(30)
Other Female Personnel	10.7%	(62)	33.1%	(81)	66.7%	(76)	86.7%	(39)
The Internet	3.8%	(22)	9.8%	(24)	12.3%	(14)	15.6%	(7)
Pamphlets	24.7%	(143)	45.7%	(112)	57.9%	(66)	84.4%	(38)
Experience with a Previous Child	2.1%	(12)	2.9%	(7)	4.4%	(5)	2.2%	(1)
Other	0.2%	(1)	1.2%	(3)	0.9%	(1)	0.0%	(0)

**Table 3 ijerph-22-00826-t003:** Infant oral hygiene practices at 6 months of age compared with the number of oral hygiene information sources consulted by the mothers.

	No Source	1 Source	2 Sources	3 Sources	4+ Sources	*p*-Value *
Gum/tooth cleaning in the past 3 months	*n* = 494	*n* = 578	*n* = 245	*n* = 114	*n* = 45	
Yes †	30.8% (152)	40.1% (232)	44.5% (109)	36.9% (42)	28.9% (13)	**0.024**
No	64.6% (319)	58.7% (339)	53.9% (132)	60.5% (69)	71.1% (32)	
Not completed §	4.6% (23)	1.2% (7)	1.6% (4)	2.6% (3)	0.0% (0)	
Frequency of teeth cleaning ^‡^	*n* = 138	*n* = 199	*n* = 93	*n* = 38	*n* = 11	
2+ times per day †	55.1% (76)	63.3% (126)	74.2% (69)	79.0% (30)	90.9% (10)	**0.002**
<2 times per day	44.9% (62)	33.7% (67)	24.7% (23)	21.0% (8)	9.1% (1)	
Not completed §	0.0% (0)	3.0% (6)	1.1% (1)	0.0% (0)	0.0% (0)	
Usage of toothpaste during cleaning ^‡^	*n* = 138	*n* = 199	*n* = 93	*n* = 38	*n* = 11	
No †	68.8% (95)	86.0% (171)	78.5% (73)	86.8% (33)	100.0% (11)	0.605
Yes	8.7% (12)	8.0% (16)	11.8% (11)	10.5% (4)	0.0% (0)	
Not completed §	22.5% (31)	6.0% (12)	9.7% (9)	2.7% (1)	0.0% (0)	

Bold values represent statistical significance. † Recommended oral hygiene behaviour with reference to Dental Care for Babies [15] and 2019 Guidelines on fluoridated toothpaste [16]. ‡ Based on respondents who responded “Yes” to the question “Have your child’s gums or teeth been cleaned in the past 3 months?” and indicated that their children had teeth at the time of the survey. § indicates respondents who did not answer the question. * Chi-squared test.

## Data Availability

The datasets presented in this article are not readily available because the data are part of an ongoing study. Requests to access the datasets should be directed to Professor Loc G. Do.

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
