# Peer review of "The Influence of Maternal Information Sources on Infant Oral Hygiene Practices for Six-Month-Olds in South Australia: A Cross-Sectional Study"

_ijerph, 2025, doi:10.3390/ijerph22060826_

Round 1

Reviewer 1 Report

Comments and Suggestions for Authors

Dear authors, the article addresses important and significant issues for the scientific community. However, it needs some adjustments before it can be published.

  1. justification and context

1.1 Can you explain more clearly why the sample was restricted to the state of South Australia? Are there demographic, epidemiological or operational reasons that make this population particularly relevant or representative?

1.2 Do the authors consider that the findings of this study can be generalized to other regions of Australia or to international contexts with similar socioeconomic characteristics?

  1. Methods and variables

2.1. The study used self-reported data from mothers. How did the authors control for or take into account social desirability or memory bias in the answers, especially with regard to hygiene practices?

2.2 The variables on hygiene practices were treated dichotomously. Did the authors consider exploring these variables as scales or continuous frequencies, which could allow for more granular analysis?

  1. Information sources - Type and quality

3.1. The authors only evaluate the number of sources consulted. Was there any attempt to assess the quality, depth or clarity of the information received by the mothers? How does this limitation affect the study's conclusions?

3.2 Would it be possible to detail which combinations of sources were most effective for positive oral hygiene practices? This would make it possible to explore potential synergistic or contradictory effects between sources.

4 Statistical analysis

4.1. Some associations showed p-values close to the significance threshold (e.g., p=0.069 for dentist). Could the authors clarify how they interpret and treat these trends in the results?

4.2 Was an interaction analysis carried out between the sources of information and sociodemographic variables (e.g. education, language, work status)? Such interactions could reveal profiles of greater impact of the intervention.

5 Discussion and practical implications

5.1. The authors mention the hypothesis of “information overload” in mothers who consulted ≥4 sources. Could there be additional data to support this hypothesis, such as measures of trust, clarity or usefulness of the information?

5.2 Considering the growing role of non-dental professionals (e.g. nurses, pharmacists), could the authors suggest interprofessional training strategies that already exist or are under development in the Australian context?

  1. Limitations and ethical considerations

6.1 In addition to the aforementioned biases, the authors considered the possibility of selection bias, since mothers who are more concerned about their health may be more likely to participate and report good practices?

6.2 Collecting data via questionnaire meant that paternal responses were excluded. Could the authors comment on the role of fathers and other caregivers, and whether there are plans to explore them in future analyses?

  1. Conclusion

7.1 The conclusion could be strengthened with practical recommendations for public health interventions based on the findings. Do the authors intend to propose more effective communication strategies, considering the results obtained?

Reviewer 2 Report

Comments and Suggestions for Authors

This article makes an important and well-supported contribution to the field of infant oral health by exploring how information sources influence mothers’ oral hygiene behaviors for their 6-month-old infants.

However, I would like to point out that the STROBE guidelines were cited as being authored by Ghaferi et al., which is incorrect. The correct citation should be:
von Elm E, Altman DG, Egger M, et al. The Strengthening the Reporting of Observational Studies in Epidemiology (STROBE) Statement: Guidelines for Reporting Observational Studies. PLoS Med. 2007.

Reviewer 3 Report

Comments and Suggestions for Authors

Dear Authors,

I read the article and I think that it is important to start to inform mothers from the early days after the  birth of the child. You  have been exemplified that informations and explications made to young mothers are important for the oral hygiene of the child. It is important the level of culture that mothers have in order to fully understand the questionnaire received . Do you have data related to this? If yes please insert them in the text.

Reviewer 4 Report

Comments and Suggestions for Authors This study evaluates the influence of maternal information sources on Infant Oral Hygiene Practices.
  1. Title should include the age of the population evaluated, as well as the type of study, to provide more information to the reader.
  2. The objectives should be written in a more concise way, placing the additional information beforehand.
  3. In the methodology section, the origin of the data should be explained more clearly, if they come from a previous study, how many phases did it include? what are the differences with this study?
  4. Can you provide any additional context to help the reader understand the origin of the data used?
  5. Can you elaborate on any concerns regarding the study's design and methodology?
  6. Are there any methodological issues or gaps that could undermine the validity or reliability of the study's findings?
  7. Could you suggest alternative approaches or improvements that would enhance the rigor of the research?
  8. As a suggestion, it would be interesting to include a figure with the main results of the study to give the reader a quick overview of them.

Round 2

Reviewer 1 Report

Comments and Suggestions for Authors

Dear authors,

The questions raised in the previous review have been effectively addressed, so the article is ready to be published.

Author Response

Dear Reviewer,

Thank you for your comments and suggestions during Revision 1 to enhance the manuscript. It is much appreciated.